# The Soliton Solutions for Some Nonlinear Fractional Differential Equations with Beta-Derivative

Erdoğan Mehmet Özkan * and Ayten Özkan

Department of Matematics, Faculty of Science, Davutpasa Campus, Yildiz Technical University, 34210 Istanbul, Turkey; uayten@yildiz.edu.tr
* Correspondence: mozkan@yildiz.edu.tr

**Abstract:** Nonlinear fractional differential equations have gained a significant place in mathematical physics. Finding the solutions to these equations has emerged as a field of study that has attracted a lot of attention lately. In this work, He's semi-inverse variation method and the ansatz method have been applied to find the soliton solutions for fractional Korteweg–de Vries equation, fractional equal width equation, and fractional modified equal width equation defined by Atangana's conformable derivative (beta-derivative). These two methods are effective methods employed to get the soliton solutions of these nonlinear equations. All of the calculations in this work have been obtained using the Maple program and the solutions have been replaced in the equations and their accuracy has been confirmed. In addition, graphics of some of the solutions are also included. The found solutions in this study have the potential to be useful in mathematical physics and engineering.

**Keywords:** He's semi inverse method; ansatz method; beta derivative

## 1. Introduction

Nonlinear partial differential equations are used to define problems in many fields of research, notably engineering. Obtaining exact solutions to such equations is a popular research topic. Fractional differential equations (FDEs) have also piqued the interest of researchers recently. Many academics have looked at FDEs in order to obtain exact answers in various methods. Many important techniques for analyzing exact solutions have been used in various research, including the ansatz method, modified simple equation method, extended trail equation, first integral, exp-function, and exp(-()) methods [1–6]. Some searchers have used alternative methods, such as the homotopy technique [7–10] and the extended Kudryashov method [11–13], modified Kudryashov and the sine-Gordon expansion approach [14–17] have also been applied by some searchers.

One of the most useful, significant, and appealing fields of study in science and engineering is Soliton's theory. Solitons are common in many aspects of life. There are often solitary observed waves that cause the soliton to emerge in shallow water on a lakeshore or in rivers. Fluid dynamics, optics, and surface wave propagation are examples of physics and engineering areas where soliton type solutions are well known. Ansatz techniques and He's Semi-Inverse method are two of the most well-known ways for getting such answers. Highly varied and intriguing soliton solutions to nonlinear equations have lately been discovered using innovative approaches [18–31].

In a wide spectrum of material science facts, Korteweg–de Vries equations have been explored as a pattern for the advancement and propagation of nonlinear waves. These equations have been presented to describe long-wavelength shallow water waves. Following that, these equations have been used to a variety of fields, including collisionless hydromagnetic waves, layered internal waves, particle acoustic waves, and plasma physics A KdV model has been used to describe a variety of speculative physical phenomena in quantum physics. In fluid dynamics and aerodynamics, it is used as a pattern for

shock wave production, solitons, turbulence, boundary layer behavior, and mass transfer [32–37]. The nonlinear time-fractional Equal Width (EW) equations are very significant partial differential equations in science and engineering that characterize a wide range of complicated nonlinear phenomena, including plasma physics, plasma waves, fluid mechanics, solid-state physics, and so on. They have derived for long waves propagating with dispersion processes. In a class of nonlinear systems, they also specify the attitude of nonlinear dispersive waves. Morrison et al. [38] proposed the equal width wave equation, which is used as a pattern in the field of fluid mechanics. Since it provides analytical solutions, this equation has inspired many scientists reading mathematical approaches for partial differential problems. Various approaches have been used to get accurate solutions for this sort of problem [39–43]. This study intends to construct soliton solutions to the time-fractional Korteweg–de Vries (KdV) equation [40,44], the time fractional equal width wave equation (EWE) and the time fractional modified fractional equal width equation (mEWE) [44] of the forms

$$\frac{\partial^\beta u}{\partial t^\beta} + au^2 u_x + bu_{xxx} = 0, \tag{1}$$

$$\frac{\partial^\beta u}{\partial t^\beta} + auu_x + b\frac{\partial^\beta}{\partial t^\beta}(u_{xx}) = 0, \tag{2}$$

$$\frac{\partial^\beta u}{\partial t^\beta} + au^2 u_x + b\frac{\partial^\beta}{\partial t^\beta}(u_{xx}) = 0, \tag{3}$$

respectively, where *a*, *b* are nonzero constants ($t > 0$, $0 < \beta \leq 1$).

## 2. Atangana's Conformable Derivatives (Beta-Derivatives) and Methodology of Solution

Conformable fractional derivatives are potentially much easier to manage, and they also follow several standard characteristics that existing fractional derivatives do not, such as the chain rule. However, this fractional derivative has a significant flaw: the fractional derivative of any differentiable function at point zero does not fulfill any physical issue and, at this time, cannot be interpreted physically. In order to extend the conformable derivative's limitation, a modified version was developed. This derivative is dependent on the interval on which the function is differentiated [45].

Abdon Atangana suggested the "beta-derivative" recently in [45–47]. The suggested version fulfills many characteristics that have been utilized to simulate various physical issues and have served as limitations for fractional derivatives. These derivatives are not fractional, but they are a natural extension of the classical derivative [48]. This derivative is dependent on the interval on which the function is differentiated. It has properties that the well-known fractional derivatives do not have, as follows.

**Definition:** *Assume that $\Psi(\omega)$ is a function. The beta derivative of $\Psi(\omega)$ is described by [45]*

$$\frac{\partial^\beta \Psi}{\partial \omega^\beta} = \lim_{\delta \to 0} \frac{\Psi\left(\omega + \delta\left(\omega + \frac{1}{\Gamma(\beta)}\right)^{1-\beta}\right) - \Psi(\omega)}{\delta}, \ \omega > 0, \ \beta \in (0,1]. \tag{4}$$

Several important properties of beta derivatives are given below [45–47].

**Theorem:** *Suppose $\Psi(\omega)$ and $\Phi(\omega)$ are $\beta$-differentiable functions for $\forall \omega > 0$ and $\beta \in (0,1]$. Then,*

- $\frac{\partial^\beta}{\partial t^\beta}\{a_0 \Psi(\omega) + a_1 \Phi(\omega)\} = a_0 \frac{\partial^\beta}{\partial t^\beta}(\Psi(\omega)) + a_1 \frac{\partial^\beta}{\partial t^\beta}(\Phi(\omega)), \ \forall a_0, a_1 \in \mathbb{R},$

- $\frac{\partial^\beta}{\partial t^\beta}(c_0) = 0, \ \forall c_0 \in \mathbb{R},$

- $\frac{\partial^\beta}{\partial t^\beta}\{\Psi(\omega)\Phi(\omega)\} = \Phi(\omega)\frac{\partial^\beta}{\partial t^\beta}\{\Psi(\omega)\} + \Psi(\omega)\frac{\partial^\beta}{\partial t^\beta}\{\Phi(\omega)\},$

- $\frac{\partial^\beta}{\partial t^\beta}\left\{\frac{\Psi(\omega)}{\Phi(\omega)}\right\} = \frac{\Phi(\omega)\frac{\partial^\beta}{\partial t^\beta}\{\Psi(\omega)\} - \Psi(\omega)\frac{\partial^\beta}{\partial t^\beta}\{\Phi(\omega)\}}{\Psi(\omega)^2},$

- $\frac{\partial^\beta}{\partial t^\beta}\{\Psi(\omega)\} = \left(\omega + \frac{1}{\Gamma(\beta)}\right)^{1-\beta}\frac{d\Psi(\omega)}{d\omega}$,

- $\frac{\partial^\beta}{\partial t^\beta}\{(\Psi \circ \Phi)(\omega)\} = \left(\omega + \frac{1}{\Gamma(\beta)}\right)^{1-\beta}\Phi\prime(\omega)\Psi\prime(\Phi(\omega))$.

The proofs of these relations are given by Atangana in [45–47].

Now, we will regard the nonlinear FDE of the type below

$$H(u, \frac{\partial^\beta u}{\partial t^\beta}, u_x, \frac{\partial^{2\beta} u}{\partial t^{2\beta}}, u_{xx}, \dots) = 0, \ \ 0 < \beta \leq 1, \tag{5}$$

where $u$ is an unknown function, $H$ is a polynomial of $u$ and its partial fractional derivatives, and $\beta$ is order of the Atangana's conformable derivatives (beta-derivatives) of the function $u = u(x, t)$.

The traveling wave transformation is

$$u(x, t) = U(\varepsilon), \ \ \varepsilon = kx - \frac{1}{\beta}\left(ct + \frac{1}{\Gamma(\beta)}\right)^\beta, \tag{6}$$

where $k \neq 0$ and $c \neq 0$ are constants. Substituting Equation (6) to Equation (5), we gain the following nonlinear ordinary differential equation (ODE)

$$N(U, \frac{dU}{d\varepsilon}, \frac{d^2 U}{d\varepsilon^2}, \frac{d^3 U}{d\varepsilon^3}, \dots) = 0. \tag{7}$$

*He's Semi-Inverse Method*

We present He's semi-inverse method (HSIM) for the exact solution of nonlinear time fractional differential equations, built by Jabbari et al. [49].

The method includes the following steps:

1.  Firstly, with the help of the above operations, Equation (5) is converted to Equation (7);
2.  If possible, Equation (7) is integrated term by term, one or more times. For convenience, the integration constant(s) can be equaled to zero;
3.  The following trial functional (8) is constructed

$$J(U) = \int L d\varepsilon \tag{8}$$

where $L$ is an unknown function of $U$ and its derivatives;

4.  By the Ritz method, different solitary wave solutions can be obtained, such as

$$U(\varepsilon) = A\text{sech}(B\varepsilon), \ \ U(\varepsilon) = A\text{csch}(B\varepsilon), \ \ U(\varepsilon) = A\tanh(B\varepsilon), \ \ U(\varepsilon) = A\coth(B\varepsilon)$$

and so on. In this study we investigated a solitary wave solution in this form

$$U(\varepsilon) = A\text{sech}(B\varepsilon), \tag{9}$$

where $A$ and $B$ are constants to be further determined. Substituting Equation (9) into Equation (8) and making $J$ stationary with respect to $A$ and $B$ results in

$$\frac{\partial J}{\partial A} = 0, \ \ \frac{\partial J}{\partial B} = 0. \tag{10}$$

By solving system (10), $A$ and $B$ are found. Thus, the solitary wave solution (9) is well-determined.

From now on, HSIM and AM will be written respectively instead of the He's semi inverse method and ansatz method, throughout the study.

### 3. Applications

*3.1. Time-Fractional Korteweg-de Vries (KdV) Equation*

To solve Equation (1), applying the traveling wave transformation (6), we gain

$$-cU' + akU^2U' + bk^3U''' = 0.$$

Integrating with respect to $\varepsilon$ once and equaling the constants of integration to zero, we get

$$-cU + \frac{ak}{3}U^3 + bk^3U'' = 0 \tag{11}$$

with $U' = \frac{dU}{d\varepsilon}$.

3.1.1. Application of HSIM

By He's semi-inverse principle [50,51], from (11), this variational formula can be found:

$$J = \int_0^\infty \left( \frac{-bk^3}{2}(U\prime)^2 - \frac{c}{2}U^2 + \frac{ak}{12}U^4 \right) d\varepsilon. \tag{12}$$

By Ritz-like method, we seek a solitary wave solution in this form

$$U(\varepsilon) = A\mathrm{sech}(B\varepsilon), \tag{13}$$

where $A$ and $B$ are unknown constants to be found later. Substituting Equation (13) into Equation (12), we have

$$J = \frac{-bk^3}{6}A^2B - \frac{c}{2B}A^2 + \frac{ak}{18B}A^4.$$

Making $J$ stationary with $A$ and $B$ gives

$$\frac{\partial J}{\partial A} = \frac{-bk^3}{3}AB - \frac{c}{B}A + \frac{2ak}{9B}A^3 = 0,$$
$$\frac{\partial J}{\partial B} = \frac{-bk^3}{6}A^2 + \frac{c}{2B^2}A^2 - \frac{ak}{18B^2}A^4 = 0.$$

From this system, we obtain

$$A = \mp\sqrt{\frac{6c}{ak}}, \quad B = \mp\sqrt{\frac{c}{bk^3}}. \tag{14}$$

Applying Equation (6), we have the following bright soliton solutions of Equation (1)

$$u(x,t) = \mp\sqrt{\frac{6c}{ak}} \ \mathrm{sech}[(\mp\sqrt{\frac{c}{bk^3}})(kx - \frac{1}{\beta}(ct + \frac{1}{\Gamma(\beta)})^\beta)]. \tag{15}$$

3.1.2. Application of AM

The bright soliton and singular soliton solutions to this equation will be found in this section. For the bright soliton solutions, we regard the ansatz

$$U(\varepsilon) = A\mathrm{sech}(\theta\varepsilon), \tag{16}$$

where

$$\varepsilon = kx - \frac{1}{\beta}(ct + \frac{1}{\Gamma(\beta)})^\beta \tag{17}$$

$A$ is the amplitude of the soliton, $\theta$ is the inverse width of the soliton and $p > 0$ is the situation for solitons to exist [52]. Now, we get

$$\frac{d^2U}{d\varepsilon^2} = Ap^2\theta^2 \ \mathrm{sech}^p(\theta\varepsilon) - Ap(p+1)\theta^2 \ \mathrm{sech}^{p+2}(\theta\varepsilon), \tag{18}$$

and

$$U^3 = A^3 \text{sech}^{3p}(\theta \varepsilon). \tag{19}$$

Substituting (16)–(19) into (11), the following equation

$$-cA\,\text{sech}^p(\theta\varepsilon) + \frac{ak}{3}A^3\text{sech}^{3p}(\theta\varepsilon) + bk^3Ap^2\theta^2\text{sech}^p(\theta\varepsilon) - bk^3\theta^2Ap(p+1)\text{sech}^{p+2}(\theta\varepsilon) = 0$$

is found.

With the aid of the balancing principle, we may get $p = 1$ by equating the exponents $p + 2$ and $3p$ in this equation. When we compare the different powers of $\text{sech}(\theta\varepsilon)$, we get the algebraic equation system shown below.

$$\frac{ak}{3}A^3 - 2bk^3\theta^2A = 0,$$
$$-cLA + bk^3\theta^2A = 0.$$

By solving this system, we obtain

$$A^{(1)} = -\sqrt{\frac{6c}{ak}}, \quad \theta^{(1)} = -\sqrt{\frac{c}{bk^3}}, \tag{20}$$

$$A^{(2)} = -\sqrt{\frac{6c}{ak}}, \quad \theta^{(2)} = \sqrt{\frac{c}{bk^3}}, \tag{21}$$

$$A^{(3)} = \sqrt{\frac{6c}{ak}}, \quad \theta^{(3)} = -\sqrt{\frac{c}{bk^3}}, \tag{22}$$

$$A^{(4)} = \sqrt{\frac{6c}{ak}}, \quad \theta^{(4)} = \sqrt{\frac{c}{bk^3}}. \tag{23}$$

Eventually, we get the bright soliton solutions for Equation (1) as follows

$$u_1(x,t) = -\sqrt{\frac{6c}{ak}}\,\text{sech}[(-\sqrt{\frac{c}{bk^3}})(kx - \frac{1}{\beta}(ct + \frac{1}{\Gamma(\beta)})^\beta)], \tag{24}$$

$$u_2(x,t) = -\sqrt{\frac{6c}{ak}}\,\text{sech}[(\sqrt{\frac{c}{bk^3}})(kx - \frac{1}{\beta}(ct + \frac{1}{\Gamma(\beta)})^\beta)], \tag{25}$$

$$u_3(x,t) = \sqrt{\frac{6c}{ak}}\,\text{sech}[(-\sqrt{\frac{c}{bk^3}})(kx - \frac{1}{\beta}(ct + \frac{1}{\Gamma(\beta)})^\beta)], \tag{26}$$

$$u_4(x,t) = \sqrt{\frac{6c}{ak}}\,\text{sech}[(\sqrt{\frac{c}{bk^3}})(kx - \frac{1}{\beta}(ct + \frac{1}{\Gamma(\beta)})^\beta)]. \tag{27}$$

For the singular soliton solutions, we regard the ansatz

$$U(\varepsilon) = A\text{sech}^p(\theta\varepsilon), \tag{28}$$

where

$$\varepsilon = kx - \frac{1}{\beta}(ct + \frac{1}{\Gamma(\beta)})^\beta \tag{29}$$

$A$ is the amplitude of the soliton, $\theta$ is the inverse width of the soliton and $p > 0$ is the situation for solitons to exist. Now, we have

$$\frac{d^2U}{d\varepsilon^2} = Ap^2\theta^2\,\text{csch}^p(\theta\varepsilon) + Ap(p+1)\theta^2\text{csch}^{p+2}(\theta\varepsilon) \tag{30}$$

and

$$U^3 = A^3\text{csch}^{3p}(\theta\varepsilon). \tag{31}$$

Substituting (28)–(31) into (11), the following equation

$$- cA \, \text{csch}^p(\theta\varepsilon) + \frac{ak}{3}A^3\text{csch}^{3p}(\theta\varepsilon) + bk^3Ap^2\theta^2\text{csch}^p(\theta\varepsilon)$$

is obtained.

In this equation, with the help of the balancing principle, equating the exponents $p + 2$ and $3p$, we get $p = 1$. Now comparing the different powers of $\text{csch}(\theta\varepsilon)$, we get the following algebraic equation system

$$\frac{ak}{3}A^3 + 2bk^3\theta^2A = 0,$$
$$-cA + bk^3\theta^2A = 0.$$

By solving this system, we find

$$A^{(1)} = -\sqrt{\frac{-6c}{ak}}, \quad \theta^{(1)} = -\sqrt{\frac{c}{bk^3}}, \tag{32}$$

$$A^{(2)} = -\sqrt{\frac{-6c}{ak}}, \quad \theta^{(2)} = \sqrt{\frac{c}{bk^3}}, \tag{33}$$

$$A^{(3)} = \sqrt{\frac{-6c}{ak}}, \quad \theta^{(3)} = -\sqrt{\frac{c}{bk^3}}, \tag{34}$$

$$A^{(4)} = \sqrt{\frac{-6c}{ak}}, \quad \theta^{(4)} = \sqrt{\frac{c}{bk^3}} \tag{35}$$

where $a < 0, \ b \neq 0$.

Consequently, we get the singular soliton solutions Equation (1) as follows

$$u_1(x,t) = -\sqrt{\frac{-6c}{ak}} \, \text{csch}[(-\sqrt{\frac{c}{bk^3}})(kx - \frac{1}{\beta}(ct + \frac{1}{\Gamma(\beta)})^\beta)], \tag{36}$$

$$u_2(x,t) = -\sqrt{\frac{-6c}{ak}} \, \text{csch}[(\sqrt{\frac{c}{bk^3}})(kx - \frac{1}{\beta}(ct + \frac{1}{\Gamma(\beta)})^\beta)], \tag{37}$$

$$u_3(x,t) = \sqrt{\frac{-6c}{ak}} \, \text{csch}[(-\sqrt{\frac{c}{bk^3}})(kx - \frac{1}{\beta}(ct + \frac{1}{\Gamma(\beta)})^\beta)], \tag{38}$$

$$u_4(x,t) = \sqrt{\frac{-6c}{ak}} \, \text{csch}[(\sqrt{\frac{c}{bk^3}})(kx - \frac{1}{\beta}(ct + \frac{1}{\Gamma(\beta)})^\beta)]. \tag{39}$$

When comparing the acquired findings to the results in [38,44], it is clear that the solutions are novel. The graphs of bright soliton solutions of $u(x,t)$ for $\beta = 0.25, 0.5$ and $0.75$ are shown in Figure 1.

Time-Fractional Korteweg-de Vries (KdV) Equation has been applied with Riemann-Liouville fractional derivative in [53].

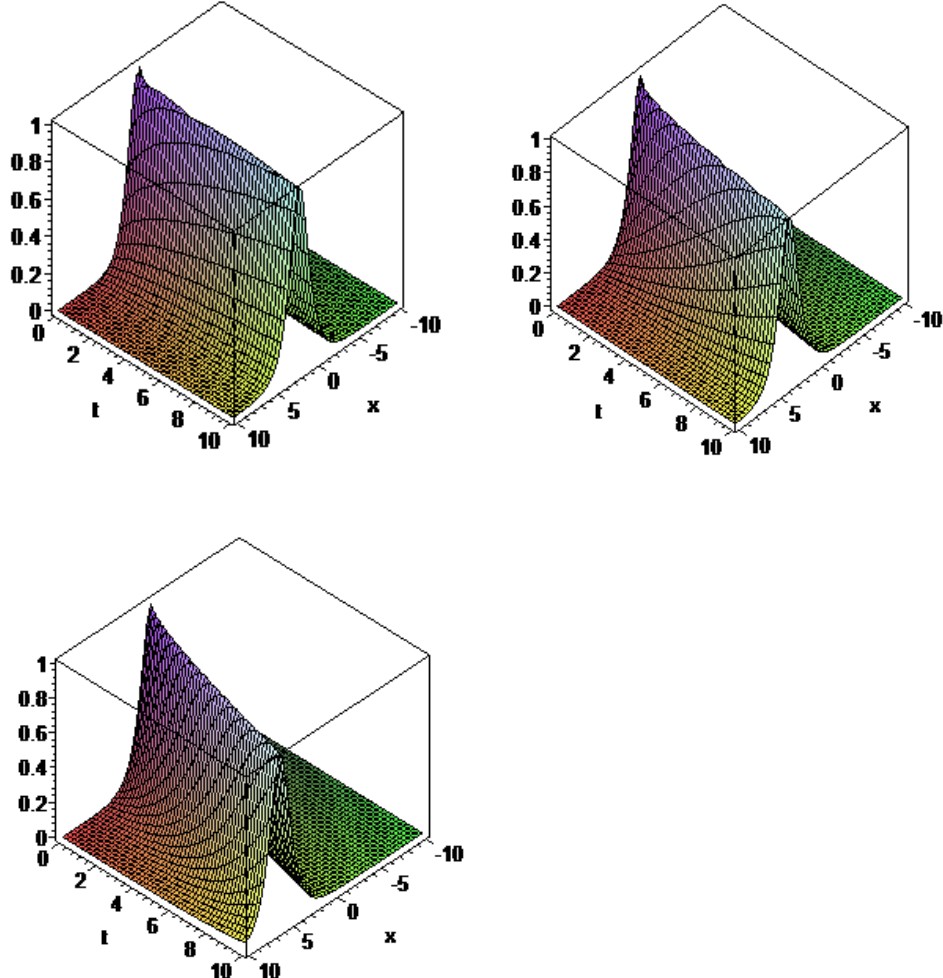

**Figure 1.** Graph of the solution $u(x, t)$ for the values $\beta = 0.25$, 0.5 and 0.75 when $a = 1$, $b = 1$, $k = c = 1$.

*3.2. Time-Fractional Equal Width Wave Equation (EWE)*

　　In the present section to solve Equation (2), using the traveling wave transformation (6), we get

$$- cU' + akUU' - bck^2U''' = 0.$$

　　Similarly, by integrating this equation and equaling the integration constants to zero, we have

$$- cU + \frac{ak}{3}U^3 + bk^3U'' = 0 \tag{40}$$

where $U' = \frac{dU}{d\varepsilon}$.

### 3.2.1. Application of HSIM

　　By He's semi-inverse principle [50,51], from (40), the variational formula can be found:

$$J = \int_0^\infty \left( \frac{bck^2}{2}(U\prime)^2 - \frac{c}{2}U^2 + \frac{ak}{6}U^3 \right) d\varepsilon. \tag{41}$$

　　By a Ritz-like method, we search for a solitary wave solution in this format

$$U(\varepsilon) = A\text{sech}^2(B\varepsilon), \tag{42}$$

where $A$ and $B$ are unknown constants. Substituting Equation (42) into Equation (41), we get

$$J = \frac{4}{15} bck^2 A^2 B - \frac{c}{3B} A^2 + \frac{4ak}{45B} A^3. \tag{43}$$

Making $J$ stationary with $A$ and $B$ gives

$$\frac{\partial J}{\partial A} = \frac{8bck^2}{15} AB - \frac{2c}{3B} A + \frac{4ak}{15B} A^2 = 0,,$$
$$\frac{\partial J}{\partial B} = \frac{4bck^2}{15} A^2 + \frac{c}{3B^2} A^2 - \frac{4ak}{45B^2} A^3 = 0.$$

From this system, we obtain

$$A = \frac{3c}{ak}, \quad B = \mp \sqrt{\frac{-1}{4bk^2}} \quad (b < 0). \tag{44}$$

Using Equation (6), we get the following bright soliton solutions of Equation (2)

$$u(x,t) = \mp \sqrt{\frac{6c}{ak}} \ \mathrm{sech}[(\mp \sqrt{\frac{c}{bk^3}})(kx - \frac{1}{\beta}(ct + \frac{1}{\Gamma(\beta)})^{\beta})]. \tag{45}$$

3.2.2. Application of AM

In this part we will achieve the bright soliton and singular soliton solutions of this equation. For the bright soliton solutions, we regard the ansatz

$$U(\varepsilon) = A\mathrm{sech}^p(\theta\varepsilon), \tag{46}$$

where

$$\varepsilon = kx - \frac{1}{\beta}(ct + \frac{1}{\Gamma(\beta)})^{\beta} \tag{47}$$

$A$ is the amplitude of the soliton, $\theta$ is the inverse width of the soliton and $p > 0$ is the condition for solitons to exist [52]. Now, we get

$$\frac{d^2 U}{d\varepsilon^2} = Ap^2\theta^2 \ \mathrm{sech}^p(\theta\varepsilon) - Ap(p+1)\theta^2 \ \mathrm{sech}^{p+2}(\theta\varepsilon), \tag{48}$$

and

$$U^2 = A^2\mathrm{sech}^{2p}(\theta\varepsilon). \tag{49}$$

Substituting (46)–(49) into (40), the following equation

$$- cA \ \mathrm{sec} \ \mathrm{h}^p(\theta\varepsilon) + \frac{ak}{2} A^2 \ \mathrm{sech}^{2p}(\theta\varepsilon) - bck^2 Ap^2\theta^2\mathrm{sech}^p(\theta\varepsilon) + bck^2\theta^2 Ap(p+1)\mathrm{sech}^{p+2}(\theta\varepsilon) = 0$$

is obtained.

By using the balancing principle to this equation, equating the exponents $p+2$ and $2p$, we get $p = 2$. When we compare the different powers of $\mathrm{sech}(\theta\varepsilon)$, we obtain the algebraic equations system presented below.

$$\frac{ak}{2} A^2 + 6bck^2\theta^2 A = 0,$$
$$-cA - 4bck^2\theta^2 A = 0.$$

Solving this system, we have

$$A = \frac{3c}{ak}, \quad \theta = \mp \sqrt{\frac{-1}{4bk^2}} \quad (b < 0). \tag{50}$$

In conclusion, we make the bright soliton solutions for Equation (2) as follows

$$u(x,t) = \frac{3c}{ak} \operatorname{sech}^2[(\mp\sqrt{\frac{-1}{4bk^2}})(kx - \frac{1}{\beta}(ct + \frac{1}{\Gamma(\beta)})^\beta)]. \tag{51}$$

For the singular soliton solutions, we take into account the ansatz

$$U(\varepsilon) = A\operatorname{csch}^p(B\varepsilon), \tag{52}$$

where

$$\varepsilon = kx - \frac{1}{\beta}(ct + \frac{1}{\Gamma(\beta)})^\beta \tag{53}$$

$A$ is the amplitude of the soliton, $\theta$ is the inverse width of the soliton and $p > 0$ is the situation for solitons to exist. Now, we have

$$\frac{d^2U}{d\varepsilon^2} = Ap^2\theta^2 \operatorname{csch}^p(\theta\varepsilon) + Ap(p+1)\theta^2 \operatorname{csch}^{p+2}(\theta\varepsilon), \tag{54}$$

and

$$U^2 = A^2\operatorname{csch}^{2p}(\theta\varepsilon). \tag{55}$$

Substituting (52)–(55) into (40), the following equation

$$-cA \operatorname{csch}^p(\theta\varepsilon) + \frac{ak}{2}A^2\operatorname{csch}^{2p}(\theta\varepsilon) - bck^2Ap^2\theta^2\operatorname{csch}^p(\theta\varepsilon) - bck^2\theta^2Ap(p+1) \operatorname{csch}^{p+2}(\theta\varepsilon) = 0$$

is obtained.

From this equation, employing the balancing principle, equating the exponents $p + 2$ and $2p$, we find $p = 2$. Now comparing the different powers of $\operatorname{csch}(\theta\varepsilon)$, we achieve the following algebraic equation system

$$\frac{ak}{2}A^2 - 6bck^2\theta^2A = 0,$$
$$-cA - 4bck^2\theta^2A = 0.$$

Solving this system, we find

$$A = \frac{-3c}{ak}, \quad \theta = \mp\sqrt{\frac{-1}{4bk^2}} \quad (b < 0). \tag{56}$$

Finally, we gain the singular soliton solutions for Equation (2) as follows

$$u(x,t) = \frac{-3c}{ak} \operatorname{csch}^2[(\mp\sqrt{\frac{-1}{4bk^2}})(kx - \frac{1}{\beta}(ct + \frac{1}{\Gamma(\beta)})^\beta)]. \tag{57}$$

when the obtained results are compared with the results in [41], it is clear that the solutions are new. In Figure 2, graphs of bright soliton solutions of $u(x,t)$ corresponding to $\beta = 0.25$, 0.5 and 0.75 are presented.

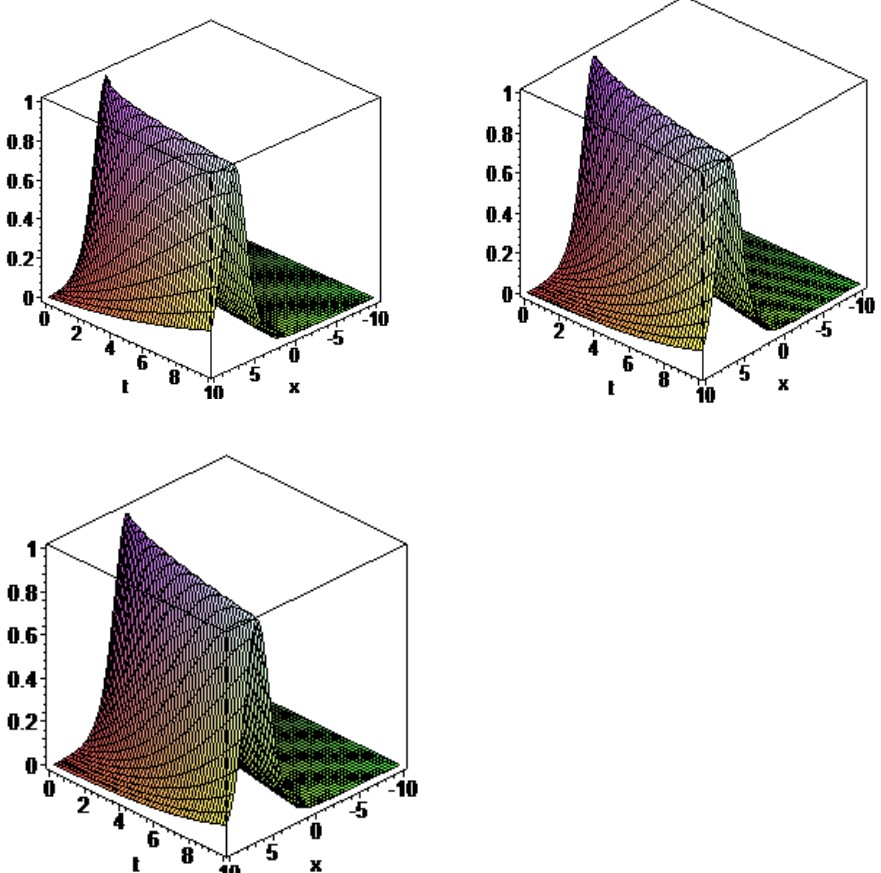

**Figure 2.** Graph of the solution $u(x, t)$ for the values $\beta$ = 0.25, 0.5 and 0.75 when $a = 3$, $b = -1$, $k = c = 1$.

*3.3. Time-Fractional Modified Equal Width Wave Equation (mEWE)*

In this section to solve Equation (3), using Equation (6), we get

$$- cU' + akU^2U' - bck^2U''' = 0.$$

In the same way, by integrating this equation and setting the integration constants to zero, we have

$$- cU + \frac{ak}{3}U^3 - bck^2U'' = 0 \tag{58}$$

where $U' = \frac{dU}{d\varepsilon}$.

3.3.1. Application of HSIM

By He's semi-inverse principle [50,51], from (58), the variational formula can be got:

$$J = \int_0^\infty \left( \frac{bck^2}{2}(U\prime)^2 - \frac{c}{2}U^2 + \frac{ak}{12}U^4 \right) d\varepsilon. \tag{59}$$

By Ritz-like method, we seek a solitary wave solution in this style

$$U(\varepsilon) = A\mathrm{sech}(B\varepsilon), \tag{60}$$

where $A$ and $B$ are unknown constants. Substituting Equation (60) into Equation (59), we find

$$J = \frac{bck^2}{6}A^2B - \frac{c}{2B}A^2 + \frac{ak}{18B}A^4. \tag{61}$$

Making $J$ stationary with $A$ and $B$ gives

$$\frac{\partial J}{\partial A} = \frac{bck^2}{3}AB - \frac{c}{B}A + \frac{2ak}{9B}A^3 = 0,$$
$$\frac{\partial J}{\partial B} = \frac{bck^2}{6}A^2 + \frac{c}{2B^2}A^2 - \frac{ak}{18B^2}A^4 = 0.$$

From this system, we obtain

$$A = \mp\sqrt{\frac{6c}{ak}}, \quad B = \mp\sqrt{\frac{-1}{bk^2}} \quad (b < 0). \tag{62}$$

Using Equation (6), we acquire the bright soliton solutions Equation (3)

$$u(x,t) = \mp\sqrt{\frac{6c}{ak}} \ \mathrm{sech}[(\mp\sqrt{\frac{-1}{bk^2}})(kx - \frac{1}{\beta}(ct + \frac{1}{\Gamma(\beta)})^\beta)] \quad (b < 0). \tag{63}$$

### 3.3.2. Application of AM

In this part, we will get the bright soliton and singular soliton solutions of this equation. For the bright soliton solutions, we allow in the ansatz

$$U(\varepsilon) = A\sec h^p(B\varepsilon), \tag{64}$$

where

$$\varepsilon = kx - \frac{1}{\beta}(ct + \frac{1}{\Gamma(\beta)})^\beta \tag{65}$$

$A$ is the amplitude of the soliton, $\theta$ is the inverse width of the soliton and $p > 0$ is for the existence of solitons [52]. Now, we get

$$\frac{d^2U}{d\varepsilon^2} = Ap^2\theta^2 \ \mathrm{sech}^p(\theta\varepsilon) - Ap(p+1)\theta^2 \ \mathrm{sech}^{p+2}(\theta\varepsilon), \tag{66}$$

and

$$U^3 = A^3\mathrm{sech}^{3p}(\theta\varepsilon). \tag{67}$$

Substituting (64)–(67) into (58), the following equation

$$-cA\sec h^p(\theta\varepsilon) + \frac{ak}{3}A^3\mathrm{sech}^{3p}(\theta\varepsilon) - bck^2Ap^2\theta^2\mathrm{sech}^p(\theta\varepsilon) + bck^2\theta^2Ap(p+1)\mathrm{sech}^{p+2}(\theta\varepsilon) = 0$$

is obtained. From this equation, by the balancing principle, equating the exponents $p+2$ and $3p$, we get $p = 1$. Now comparing the different powers of $\mathrm{sech}(\theta\varepsilon)$, we achieve the following algebraic equation system

$$\frac{ak}{3}A^3 + 2bck^2\theta^2A = 0,$$
$$-cA - bck^2\theta^2A = 0.$$

Solving this system, we get

$$A^{(1)} = -\sqrt{\frac{6c}{ak}}, \quad \theta^{(1)} = -\sqrt{\frac{-1}{bk^2}}, \tag{68}$$

$$A^{(2)} = -\sqrt{\frac{6c}{ak}}, \quad \theta^{(2)} = \sqrt{\frac{-1}{bk^2}}, \tag{69}$$

$$A^{(3)} = \sqrt{\frac{6c}{ak}}, \quad \theta^{(3)} = -\sqrt{\frac{-1}{bk^2}}, \tag{70}$$

$$A^{(4)} = \sqrt{\frac{6c}{ak}}, \quad \theta^{(4)} = \sqrt{\frac{-1}{bk^2}}. \tag{71}$$

In conclusion, we get the bright soliton solutions for Equation (3) as follows

$$u_1(x,t) = -\sqrt{\frac{6c}{ak}}\ \text{sech}[(-\sqrt{\frac{-1}{bk^2}})(kx - \frac{1}{\beta}(ct + \frac{1}{\Gamma(\beta)})^\beta)],$$

(72)

$$u_2(x,t) = -\sqrt{\frac{6c}{ak}}\ \text{sech}[(\sqrt{\frac{-1}{bk^2}})(kx - \frac{1}{\beta}(ct + \frac{1}{\Gamma(\beta)})^\beta],$$

(73)

$$u_3(x,t) = \sqrt{\frac{6c}{ak}}\ \text{sech}[(-\sqrt{\frac{-1}{bk^2}})(kx - \frac{1}{\beta}(ct + \frac{1}{\Gamma(\beta)})^\beta)],$$

(74)

$$u_4(x,t) = \sqrt{\frac{6c}{ak}}\ \text{sech}[(\sqrt{\frac{-1}{bk^2}})(kx - \frac{1}{\beta}(ct + \frac{1}{\Gamma(\beta)})^\beta)].$$

(75)

For the singular soliton solutions, we regard the ansatz

$$U(\varepsilon) = A\text{csch}^p(\theta\varepsilon),$$

(76)

where

$$\varepsilon = kx - \frac{1}{\beta}(ct + \frac{1}{\Gamma(\beta)})^\beta$$

(77)

$A$ is the amplitude of the soliton, $\theta$ is the inverse width of the soliton and $p > 0$ is for the existence solitons, as well. Now, we have

$$\frac{d^2U}{d\varepsilon^2} = Ap^2\theta^2\ \text{csch}^p(\theta\varepsilon) + Ap(p+1)\theta^2\ \text{csch}^{p+2}(\theta\varepsilon),$$

(78)

and

$$U^3 = A^3\text{csch}^{3p}(\theta\varepsilon).$$

(79)

Substituting (76)–(79) into (58), the following equation

$$-cA\ \text{csch}^p(\theta\varepsilon) + \frac{ak}{3}A^3\text{csch}^{3p}(\theta\varepsilon) - bck^2Ap^2\theta^2\text{csch}^p(\theta\varepsilon)$$

is obtained. From this equation, by the balancing principle, equating the exponents $p + 2$ and $3p$, we get $p = 1$. Now comparing the different powers of $\text{csch}(\theta\varepsilon)$, we find the following algebraic system

$$\frac{ak}{3}A^3 - 2bck^2\theta^2A = 0,$$
$$-cA - bck^2\theta^2A = 0.$$

Solving this system, we get

$$A^{(1)} = -\sqrt{\frac{-6c}{ak}},\quad \theta^{(1)} = -\sqrt{\frac{-1}{bk^2}},$$

(80)

$$A^{(2)} = -\sqrt{\frac{-6c}{ak}},\quad \theta^{(2)} = \sqrt{\frac{-1}{bk^2}},$$

(81)

$$A^{(3)} = \sqrt{\frac{-6c}{ak}},\quad \theta^{(3)} = -\sqrt{\frac{-1}{bk^2}},$$

(82)

$$A^{(4)} = \sqrt{\frac{-6c}{ak}},\quad \theta^{(4)} = \sqrt{\frac{-1}{bk^2}}.$$

(83)

Eventually, we gain the singular soliton solutions for Equation (3) as follows

$$u_1(x,t) = -\sqrt{\frac{-6c}{ak}}\ \text{csch}[(-\sqrt{\frac{-1}{bk^2}})(kx - \frac{1}{\beta}(ct + \frac{1}{\Gamma(\beta)})^\beta)],$$

(84)

$$u_2(x,t) = -\sqrt{\frac{-6c}{ak}} \ \mathrm{csch}[(\sqrt{\frac{-1}{bk^2}})(kx - \frac{1}{\beta}(ct + \frac{1}{\Gamma(\beta)})^{\beta})], \tag{85}$$

$$u_3(x,t) = \sqrt{\frac{-6c}{ak}} \ \mathrm{csch}[(-\sqrt{\frac{-1}{bk^2}})(kx - \frac{1}{\beta}(ct + \frac{1}{\Gamma(\beta)})^{\beta})], \tag{86}$$

$$u_4(x,t) = \sqrt{\frac{-6c}{ak}} \ \mathrm{csch}[(\sqrt{\frac{-1}{bk^2}})(kx - \frac{1}{\beta}(ct + \frac{1}{\Gamma(\beta)})^{\beta})]. \tag{87}$$

when the findings obtained are compared with the results in [41], it is clear that the solutions are new.

On bright soliton solution, if $a = c = k = 1$ and $b = -1$, it is clear that the graphs in Figure 1 will be the same.

## 4. Conclusions

In this work, He's semi-inverse variation method and the ansatz method have been used successfully to obtain the bright and singular soliton solutions of the nonlinear fractional KdV equation, EWE and mEWE. It could be deduced from the findings that these techniques are suited. It is understood that the other soliton solutions can be obtained with them. They can be considered more powerful and convenient in solving other nonlinear FDE types. The resulting soliton solutions are useful to researchers and have important applications in mathematical physics and engineering. By selecting the appropriate parameter values, the behaviors of several solutions have been given that aid the investigator in understanding the physical comment. In addition, when the results obtained by both methods are compared with related publications, it is seen that they are new. It is understood from the results gained that the proposed techniques are so effective, promising, and suitable for solving other nonlinear fractional differential equations.

**Author Contributions:** The authors E.M.Ö. and A.Ö. contributed equally to this work. All authors have read and agreed to the published version of the manuscript.

**Funding:** This research received no external funding.

**Data Availability Statement:** Not applicable.

**Acknowledgments:** The authors are grateful to the referee for his helpful recommendations, which helped mold the article into what it is now.

**Conflicts of Interest:** The authors declare no conflict of interest.

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
