# Peer review of "The Soliton Solutions for Some Nonlinear Fractional Differential Equations with Beta-Derivative"

_axioms, doi:10.3390/axioms10030203_

Round 1

Reviewer 1 Report

To connect fractional calculus, perturbation method and solitons would be nice and important in applied mathematics and the research on which the paper is written might be an important step in this direction. Unfortunately, the present form of the paper cannot be accepted. Details are in attached files.

Author Response

Dear reviewer,

You will find the report with the explanations attached. We believe that we act according to all the statements you make. We are so grateful for everything.

Best regards

Reviewer 2 Report

The authors apply semi-inverse variation method and the ansatz method to find  the soliton  solutions for fractional ] Korteverg de Vries equation (KdV equation ). All calculations are obtained by using the Maple program, .  The paper is well written and well organized .The considered problem  belong to the computer science methods in numerical analysis. The paper don't contain any new mathematical result. It is rather interesting only from the computer science point of view.

Author Response

Dear reviewer,

You will find the article with the corrections attached. We believe that we act according to all the statements you make. We are so grateful for everything.

Best regards

Reviewer 3 Report

See attached file.

Author Response

Dear reviewer,

You will find the report with the explanations attached. We believe that we act according to all the statements you make. We are so grateful for everything.

Best regards

The paper needs a deep revision since there are some obscure technical results and some typos to be fixed. It is also necessary to explain clearly the novelty of the presented results related to the mentioned literature. Is this only a numerically tested “confirmation” of previous results through examples?

- It is a study in which the solutions of 3 equations with beta derivatives are shown in 2 different ways. Since the results of the beta-derivative work of these equations have not been done before, we do not think that this is a confirmation of the previous results.

Some of the changes of variables and later related calculations are manipulated without clear comments or further derivation. Some more specific comments follow below:

Section 2, line 1 : The ( missed) beta derivative

- It is done

Last line of page 2: “can be got”

- It is done

Line 3 after (4): What is the meaning of the superscript “A” in the right-hand-side?

-This is typo. It is done

After the above group of equations: Delete “of” in “Taking account of”.

-Some changes have been made in this section and the "taking account of" has been removed.

After the above line: Explain why and why “delta” is equalized to the expression given, so that it depends on omega (which is a variable in (4) where delta tends to zero. This expression for “delta” as given cannot be understood, in principle, by the mentioned reasons. Extend the calculations at the end of page 2 to get the first mathematical expression of page 3.

-The definition and attributes of the beta derivative are listed in this section, as in the previous articles. As you know, we have just given them in general terms, and we have provided references with evidence and explanations for each. We've put them to work in three different equations. To minimize confusion, we've also deleted some expressions and modified them to make them more obvious.

Eqn. 60: Are A, B real constants? Please, specify.

- They are constants to be determined later according to He's semi-inverse method. According to this method, A and B will be found depending on the parameters a, b and c. Depending on the value they will take, these will be real numbers. There are similar cases in Section 3.1.1 and Section 3.2.1.

Eqn. 59: Can “c “ and “ (a*k) “to have distinct sign?

- According to Ref. [49], by He’s semi-inverse method [50,51], we can arrive at the following variational formulation. Eq. (12) and Eq. (41) are also revealed in the same way.

Reference 32, Paper title: on-> On

- It is done

There are authors´ surnames in some of the titles in the list of references with have low case initials.

- It is done

Round 2

Reviewer 1 Report

Correct paper. 

Reviewer 3 Report

The paper has beein improved with respect to its previous version.